# Don't put all social network sites in one basket: Facebook, Instagram, Twitter, TikTok, and their relations with well-being during the COVID-19 pandemic

**Alexandra Masciantonio**[1]*, **David Bourguignon**[1], **Pierre Bouchat**[1], **Manon Balty**[1], **Bernard Rimé**[2]

**1** Department of Psychology, Université de Lorraine, Metz, France, **2** Department of Psychology, Université Catholique de Louvain, Louvain-la-Neuve, Belgium

* Alexandra.masciantonio@univ-lorraine.fr

## Abstract

Prior studies indicated that actively using social network sites (SNSs) is positively associated with well-being by enhancing social support and feelings of connectedness. Conversely, passively using SNSs is negatively associated with well-being by fostering upward social comparison and envy. However, the majority of these studies has focused on Facebook. The present research examined the relationships between well-being—satisfaction with life, negative affect, positive affect—and using actively or passively various SNSs—Facebook, Instagram, Twitter, TikTok—during the COVID-19 pandemic. In addition, two mediators were tested: social support and upward social comparison. One thousand four persons completed an online survey during the quarantine measures; the analyses employed structural equation modeling. Results showed that passive usage of Facebook is negatively related to well-being through upward social comparison, whereas active usage of Instagram is positively related to satisfaction with life and negative affect through social support. Furthermore, active usage of Twitter was positively related to satisfaction with life through social support; while passive usage was negatively related to upward social comparison, which, in turn, was associated with more negative affect. Finally, TikTok use was not associated with well-being. Results are discussed in line with SNSs' architectures and users' motivations. Future research is required to go beyond methodological and statistical limitations and allow generalization. This study concludes that SNSs must be differentiated to truly understand how they shape human interactions.

## Introduction

The COVID-19 pandemic that has hit the world since the end of 2019 has led the governments of many countries to impose quarantine measures on their populations. For many people, these confinement measures led to a drastic reduction in interpersonal relations. However,

**Data Availability Statement:** Data are available in OSF (Open Science Framework) at: https://doi.org/10.17605/OSF.IO/S5MJX.

**Funding:** The author(s) received no specific funding for this work.

**Competing interests:** The authors have declared that no competing interests exist.

interpersonal relations have powerful beneficial effects on physical and mental health [1,2]. In order to cope with the negative effects of social isolation on well-being, a significant number of recommendations were issued [3,4]. Several of them, derived predominantly from non-scholars, have promoted the use of social network sites (SNSs) to keep contact with family and friends [5,6]. Nevertheless, this assertion involves addressing one complex question: are SNSs really beneficial to well-being? The present study will contribute to this research question, through a short literature review and an empirical study. More research will be required to provide a clear understanding of how SNSs impact well-being.

## Definition of key concepts

Before addressing the relationship between SNSs and well-being, both need to be defined. On one side, the familiar definition of Ellison and boyd [7 p157] described SNSs as networked communication platforms in which participants 1) have uniquely identifiable profiles that consist of user-supplied content, content provided by other users, and/or system-level data; 2) can publicly articulate connections that can be viewed and traversed by others; and 3) can consume, produce, and/or interact with streams of user-generated content provided by their connections on the site.

On the other side, the term well-being refers, in this study, to the subjective part of well-being. Instead of relying on physical and material resources, subjective well-being can be understood as "people's overall evaluations of their lives and their emotional experiences." [8 p87]. Hence, subjective well-being is a multidimensionality construct, and each component needs to be assessed individually. Typically, subjective well-being comprises at least three components: a sense of satisfaction with life, the presence of positive affect, and the absence of negative affect [8]. The satisfaction with life allows to capture how people evaluate their lives (i.e., cognitive level of subjective well-being). Likewise, positive and negative affect reflect what feelings people experience in their lives (i.e., affective level of subjective well-being).

## Literature review

The relation between SNSs and well-being may at first seem inconsistent. Several studies showed that SNSs use is negatively associated with well-being [9,10], while others revealed a positive relationship [11,12]. However, these studies relied on an overall measure of SNSs use, whereas two distinguish usages can be proposed: an active (e.g., interacting directly with others by posting content or commenting others' content) and a passive one (e.g., reading and consuming others' content). Gerson, Plagnol and Corr [13] pointed out how these usages match with specific SNS activities, demonstrating they reflect related, but separate constructs. In that respect, results seem more uniform when different modalities of SNSs use have been taken into account: actively using SNSs is positively associated with well-being and in contrast, passively using SNSs is negatively associated with well-being [14–19]. Verduyn, Ybarra, Résibois, Jonides and Kross [20] reviewed the literature and identified the mechanisms underlying these relationships. Their model suggests that actively using SNSs increases subjective well-being by improving social capital and feelings of connectedness. Conversely, passively using SNSs lessens subjective well-being by fostering social comparison and envy. Although this model is a major step to clarify the consequences of SNSs on well-being, most of the studies underpinning these mechanisms have focused on Facebook. Doing so, one can wonder whether other SNSs might have different impacts on well-being.

Few studies have investigated the effects of different SNSs on well-being. Pittman and Reich [21] demonstrated that the use of image-based platforms (e.g., Instagram, Snapchat) was positively associated with well-being and negatively with loneliness, whereas text-based platforms

(e.g., Twitter, Yik Yak) were not related to well-being and loneliness. Recently, Chae [22] has also examined the relationships between various platforms and well-being through social comparison. As expected, social comparison was negatively associated with well-being; but while Instagram and LinkedIn enhanced social comparison, Twitter decreased it. Surprisingly, Facebook use was not related to well-being. These two studies have therefore yielded contradictory outcomes, but they employed an overall measure of SNSs use which makes impossible to investigate the distinct effects of passive and active usages.

## Overview of the research

To draw conclusions on SNSs and well-being, the literature on passive and active usage need to be integrated with the literature on cross-media studies. To that end, the present research examines the relationships between various SNSs and well-being through two mediators—social support and upward social comparison. Specifically, this study focuses on the active and passive usages of four popular SNSs: Facebook, Instagram, Twitter and TikTok [23]. Although Facebook, Instagram and Twitter are henceforth well studied in the literature, TikTok is a new SNS created in 2016 with a number of users increasing day by day [24]. These SNSs differ from each other by their architectures [25]: Facebook incorporates both image and text, Twitter is text-based, and Instagram as well as TikTok are image-based (the first concerns pictures and the second relies on videos). Instagram, TikTok and Twitter are also unidirectional (i.e., possibility to follow someone's content without their approval), whereas Facebook is dyadic (i.e., need to be approved by someone to access their content). Moreover, people do not use them for the same reasons: Facebook use is mainly related to social support and self-presentation [26]; Instagram allows users to self-document, self-promote, express one's creativity and see other's content [27]; Twitter use is mainly driven by informational needs [28,29]. Finally, only one study examined TikTok use and concluded that the platform was seen as a "recording tool rather than a social media app" [30] p132]. Indeed, self-document was the most important motivation to use TikTok.

The model of Verduyn et al. [20] is mainly based on Facebook use, one would therefore expect to draw the same conclusions as the authors:

> Hypothesis 1: Social support mediates the positive association between actively using Facebook and subjective well-being, and upward social comparison mediates the negative association between passively using Facebook and subjective well-being.

Image-based SNSs, such as Instagram and TikTok, have been shown to be related to well-being [21]. Moreover, Instagram users want to keep in touch with their friends, but also to self-promote [27]. Hence, social support and upward social comparison could both play a part in this relation. One would therefore expect the model of Verduyn et al. [20] to be generalized to Instagram:

> Hypothesis 2: Social support mediates the positive association between actively using Instagram and subjective well-being, and upward social comparison mediates the negative association between passively using Instagram and subjective well-being.

In contrast, TikTok use was not firstly motivated by social interaction or self-presentation [30]. So, no assumption can be made about the mediating roles of social support and upward social comparison. The only hypothesis which can be proposed is:

Hypothesis 3: Actively using TikTok is positively associated with well-being and passively using TikTok is negatively associated with well-being.

Finally, text-based SNSs do not appear to be related to well-being [21]. The following hypothesis is therefore proposed for Twitter:

Hypothesis 4: Actively and passively using Twitter is not associated with well-being.

## Method

### Participants and procedure

One thousand four persons agreed to participate in the study. Among them, were excluded those reporting missing data and under the age of 18. The final sample was composed of 793 participants (613 women, 178 men and 2 persons who have a gender identity other than male or female) aged between 18 and 77 years old ($M = 33.75$, $SD = 14.70$). All participants were francophone: 463 were French, 264 were Belgian, 20 were Swiss and 46 had another nationality. Regarding the highest degree completed, one person had no primary education, 253 had a high school degree, 285 had a university short cycle degree (two or three years), 207 had a university long cycle degree (four or five years) and 47 had a doctorate. Finally, 89% had a Facebook account ($N = 703$), 63% had an Instagram account ($N = 502$), 38% had a Twitter account ($N = 300$) and 15% had a TikTok account ($N = 121$). An anonymous online survey was created using the Qualtrics Survey Software. Participants were recruited through academic mailing lists from social science, which explains the large proportion of women and academic people in the sample. Before completing the measures, all participants signed an informed consent form and accepted voluntary to take part in this research. Data collection was carried out from 7th April 2020 to 16th April 2020. Measures reported in the present study are part of a larger questionnaire; all data are available in OSF (Open Science Framework) at: https://osf.io/s5mjx/.

### Measures

Overall SNS use: When participants declared to have an account for one of the four SNSs (Facebook, Instagram, Twitter, TikTok), they indicated the frequency they used this SNS before and during the quarantine measures on a 7-point scale (never; between one and three times a year; less than once a month; one to four times a month; one to four times a week; one to three times a day; more than three times a day).

Active and passive usage of SNSs: To be consistent with the literature [13], we chose to measure passive and active usage as separate constructs. This means that users can have both an active and a passive SNS usage; they can spend most of their time scrolling their news feed, but they can also send messages throughout the day. When participants declared to have an account for a SNS (Facebook, Instagram, Twitter, TikTok), they were therefore asked how much they used this SNS actively (1 = not actively at all; 7 = very actively) and passively (1 = not passively at all; 7 = very passively) during the quarantine measures [19]. Active usage was defined as "posting and commenting on [Facebook][Instagram][Twitter][TikTok], for example: post content on your profile, react to posts and comments from other users, etc.", while passive usage as "browsing [Facebook][Instagram][Twitter][TikTok], for example: scrolling through your news feed, looking at other users' profiles, etc.".

Motivations to use SNSs: Three motivations to use SNSs were derived from Cheung, Chiu and Lee [31]: maintaining interpersonal interconnectivity ("To stay in touch"; "To have something to do with others"), purposive value ("To get information"; "To provide others with

information") and entertainment value ("To pass time away when bored"; "To be entertained"). Participants were asked to rate the extent to which these 6 items correspond to their motivations to use SNSs during the quarantine measures on a 7-point Likert scale (1 = strongly disagree; 7 = strongly agree). McDonald's ω computed a value of.80 for maintaining interpersonal interconnectivity,.57 for purposive value and.79 for entertainment value.

Social support on SNSs: Social support on SNSs was measured using eight items adapted from Nick et al. [32]. Two items were chosen for each subscale (emotional support, informational support, social companionship and instrumental support). Participants indicated their agreement with these items on a 7-point Likert scale. Sample items include "During quarantine measures, people show that they care about me on social network sites." and "During quarantine measures, people give me useful advice on social network sites.". Scores for each subscale were averaged such that a higher overall score indicated greater social support on SNSs during the quarantine measures (McDonald's ω = .83).

Upward social comparison: The upward social comparison was inspired from Brunot and Juhel [33] and consisted in two items: "On social network sites, I sometimes think that my relatives (friends, family and colleagues) are fare better than me during the quarantine measures" and "On social network sites, I sometimes think that my relatives (friends, family and colleagues) are better off than me". Participants indicated their agreement with these items on a 7-point Likert scale (McDonald's ω = .84).

Positive affect: Positive affect were assessed by asking participants to rate of much they feel "Optimistic, encouraged, hopeful" and "Proud, trustful, self-confident" on a 7-point Likert scale (McDonald's ω = .76). The measure was adapted from Fredrickson [34].

Negative affect: Negative affect were assessed by asking participants to rate of much they feel "Sad, depressed, unhappy", "Angry, furious" and "Anxious, frightened" on a 7-point Likert scale (McDonald's ω = .75). The measure was adapted from Gaudreau, Sanchez and Blondin [35].

Satisfaction with life: Satisfaction with life was measured using the Satisfaction with Life Scale [36]. An example item is "I am satisfied with my life". Participants indicated their agreement with the five items on a 7-point Likert scale. Given the good reliability (McDonald's ω = .89), the five items were aggregated.

## Results

Analyses were conducted using the JASP software [37].

### Preliminary analyses

An exploratory analysis of the data is available in OSF at: https://osf.io/fe4pn/. Four paired samples T-Tests have been also carried out between the overall SNS use before the quarantine measures and during the quarantine measures. Results showed that the overall use have increased during the quarantine for all SNSs, and in particular for TikTok: Facebook ($t_{(702)}$ = 11.84, $p < .001$, d = .45), Instagram ($t_{(501)}$ = 6.33, $p < .001$, d = .28), Twitter ($t_{(299)}$ = 4.02, $p < .001$, d = .23) and TikTok ($t_{(120)}$ = 10.31, $p < .001$, d = .94). Finally, correlations between overall SNS use during quarantine and motivations to use SNSs are presented in Table 1.

### Main analyses

Structural equation modeling with Lavaan [38] was used to examine the relationships between SNSs (Facebook, Instagram, Twitter, TikTok) and well-being (positive affect, negative affect and satisfaction with life) through two mediators, social support and upward social comparison. For each model, dependent variables were controlled for age and gender. A one step

**Table 1. Pearson's correlations between overall SNSs use during the quarantine measures and motivations to use SNSs.**

| | Maintaining interpersonal interconnectivity | Purposive value | Entertainment value |
|---|---|---|---|
| Overall **Facebook** use during quarantine | .101** | .076* | .135*** |
| Overall **Instagram** use during quarantine | .100* | .062 | .376*** |
| Overall **Twitter** use during quarantine | .008 | .147* | .307*** |
| Overall **TikTok** use during quarantine | .132 | .090 | .229* |

Note.

$^*p < .05$;

$^{**}p < .01$;

$^{***}p < .001$.

approach was employed, that means that the parameters of the measurement model and the structural model were estimated simultaneously. Analyses were carried on with DWLS (diagonally weighted least squares) estimator which is adapted for data violating normality [39]. Five fit indices were chosen: $\chi^2$ (chi-square), SRMR (Standard Root Mean Square Residuals), RMSEA (Root Mean Square Error of Approximation), CFI (Comparative Fit Index) and TLI (Tucker-Lewis Index) [39]. The first two address the global fit of the model: $\chi^2$ must be nonsignificant and the value of SRMR must be equal or lower to .08. RMSEA concerns the parsimony of the model and must be lower to .06. Lastly, CFI and TLI are incremental indices and must be superior to .9.

**Facebook.** All standardized item loadings exceeded .4 and were significant ($p < .001$). The results also revealed a satisfactory model fit to the data: $\chi^2(227, N = 703) = 723.86, p < .001$; SRMR = .06; RMSEA = .056; CFI = .948; TLI = .938. Although the $\chi^2$ is significant, this statistic is very sensitive to sample size [39].

As shown in Fig 1, direct paths from actively and passively using Facebook to satisfaction with life, positive affect and negative affect were nonsignificant ($p > .05$), except the path from using actively Facebook to negative affect ($\beta = .17, p < .05$). Contrary to hypothesis 1, direct path from actively using Facebook to social support was nonsignificant ($p > .05$), but direct path from passively using Facebook to upward social comparison was significant ($\beta = .13, p < .05$). All estimated paths from social support and upward social comparison to the three constructs of well-being were significant ($p < .05$). Consistent with hypothesis 1, the indirect effects of passively using Facebook on well-being (satisfaction with life, positive affect and negative affect) through upward social comparison were significant ($p < .05$).

In other words, results revealed that upward social comparison mediates the negative association between passively using Facebook and subjective well-being. Nonetheless, using actively Facebook was also directly associated with greater negative affect.

**Instagram.** All standardized items loadings exceeded .4 and were significant ($p < .001$). The results also revealed a satisfactory model fit to the data: $\chi^2(227, N = 502) = 532.32, p < .001$; SRMR = .061; RMSEA = .052; CFI = .955; TLI = .947.

As shown in Fig 2, direct paths from actively and passively using Instagram to satisfaction with life, positive affect and negative affect were nonsignificant ($p > .05$). Contrary to hypothesis 2, direct path from passively using Instagram to upward social comparison was nonsignificant ($p > .05$), but direct path from actively using Instagram to social support was significant ($\beta = .21, p < .05$). All estimated paths from social support and social comparison to the three constructs of well-being were significant ($p < .05$), except the path from social support to positive affect which was nonsignificant ($p > .05$). Partially consistent with hypothesis 2, the

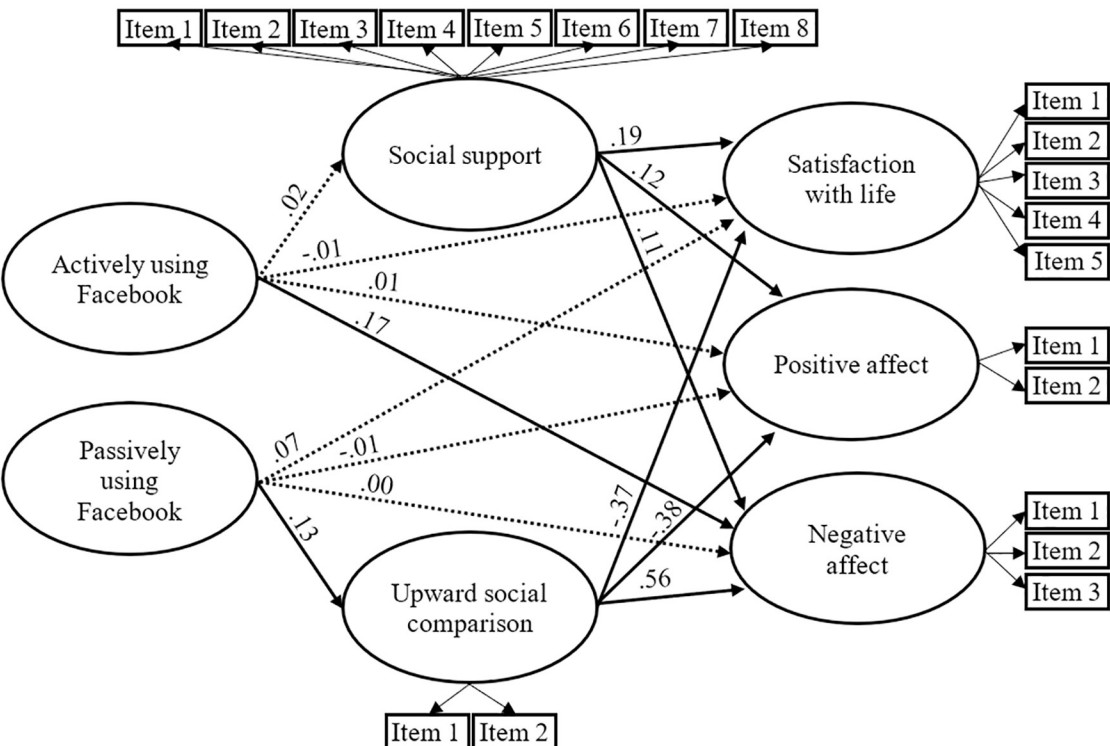

**Fig 1. The estimated standardized parameters of the Facebook model.** Dashed lined indicate nonsignificant paths ($p >.05$). The three components of well-being were controlled–but not displayed—for gender and age: Age was associated with satisfaction with life ($\beta = .26$, $p < .05$), negative affect ($\beta = -.33$, $p < .05$), and positive affect ($\beta = .20$, $p < .05$); women had less positive affect ($\beta = -.18$, $p < .05$), and more negative affect ($\beta = .12$, $p < .05$).

indirect effects of actively using Instagram on satisfaction with life and negative affect through social support was significant ($p < .05$).

In other words, results revealed that social support mediates the positive association between actively using Instagram and satisfaction with life on one hand, and the positive association between actively using Instagram and negative affect on the other.

**Twitter.** All standardized items loadings exceeded.4 and were significant ($p < .001$). The results also revealed a satisfactory model fit to the data: $\chi^2(227, N = 300) = 415.61$, $p < .001$; SRMR = .071; RMSEA = .053; CFI = .953; TLI = .944.

As shown in Fig 3, direct paths from actively and passively using Twitter to satisfaction with life, positive affect and negative affect were nonsignificant ($p >.05$). Contrary to hypothesis 4, direct path from passively using Twitter to upward social comparison was significant ($\beta = -.14$, $p < .05$), and direct path from actively using Twitter to social support was also significant ($\beta = .15$, $p < .05$). All estimated paths from social support and social comparison to the three constructs of well-being were significant ($p < .05$), except the path from social support to positive affect which was nonsignificant ($p >.05$). The indirect effect of actively using Twitter on satisfaction with life through social support was significant ($p < .05$). Likewise, the indirect effects of passively using Twitter on negative affect through upward social comparison was significant ($p < .05$), and the indirect effects of passively using Twitter on satisfaction with life and positive affect through upward social comparison was significant were near significant ($p = .06$ for satisfaction with life; $p = .056$ for positive affect).

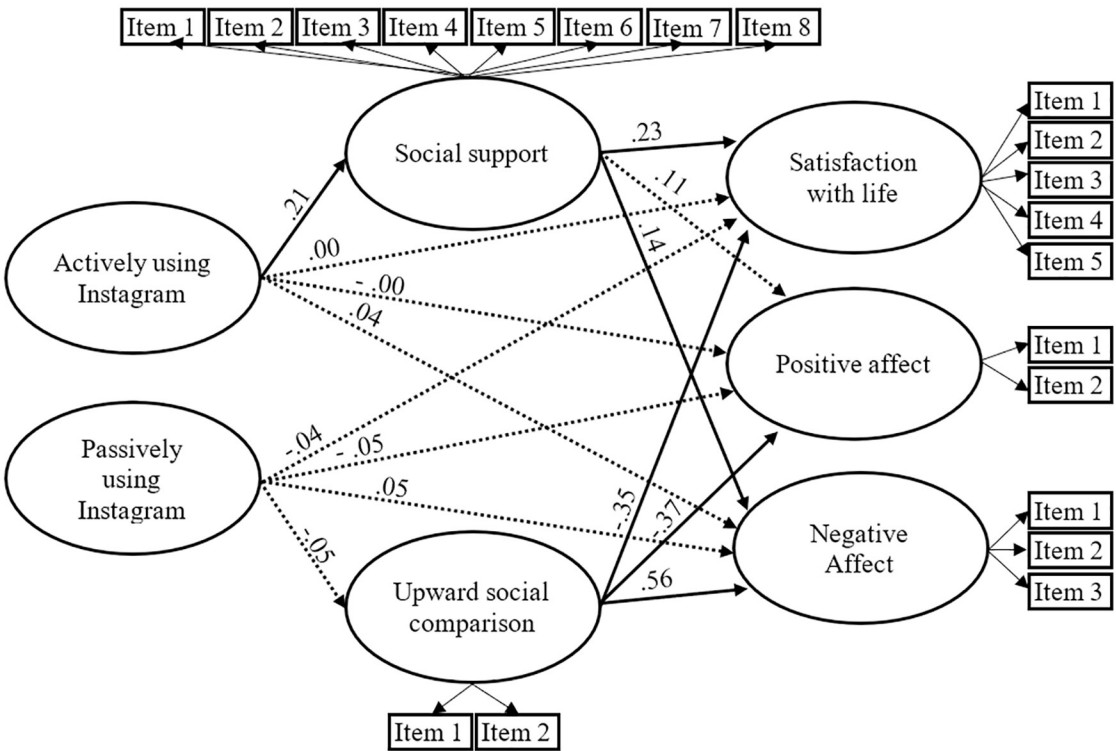

**Fig 2. The estimated standardized parameters of the Instagram model.** Dashed lined indicate nonsignificant paths (*p* >.05). The three components of well-being were controlled–but not displayed—for gender and age: Age was associated with satisfaction with life (β = .27, *p* < .05), negative affect (β = -.28, *p* < .05), and positive affect (β = .27, *p* < .05); women had less positive affect (β = -.18, *p* < .05), and more negative affect (β = .11, *p* < .05).

In other words, results showed that actively using Twitter was associated with more social support, and that using passively Twitter was associated with less upward social comparison. In addition, social support mediated the relation between using actively Twitter and satisfaction with life, and social comparison mediated the relation between passively using Twitter and negative affect.

**TikTok.**   All standardized items loadings were significant (*p* < .05) and exceeded.4, except one item of the social support construct (.24). The results also revealed that the model fits the data well: $\chi^2$(227, N = 121) = 227.034, *p* >.05; SRMR = .084; RMSEA = .001; CFI = 1.000; TLI = 1.000.

As shown in Fig 4 and inconsistent with hypothesis 3, direct paths from actively and passively using TikTok to satisfaction with life, positive affect and negative affect were nonsignificant (*p* >.05). Besides, direct paths from passively and actively using Twitter to upward social comparison and social support respectively, were nonsignificant (*p* >.05). Lastly, only paths from upward social comparison to positive and negative affect were significant (*p* < .05).

In other words, results revealed that actively and passively using Tiktok was not associated with well-being, and that social support and upward social comparison did not appear to play a meditational role between TikTok use and well-being.

## Discussion

Past researches have shown that actively using SNSs is positively associated with well-being through social support, and that passively using SNSs is negatively associated with well-being

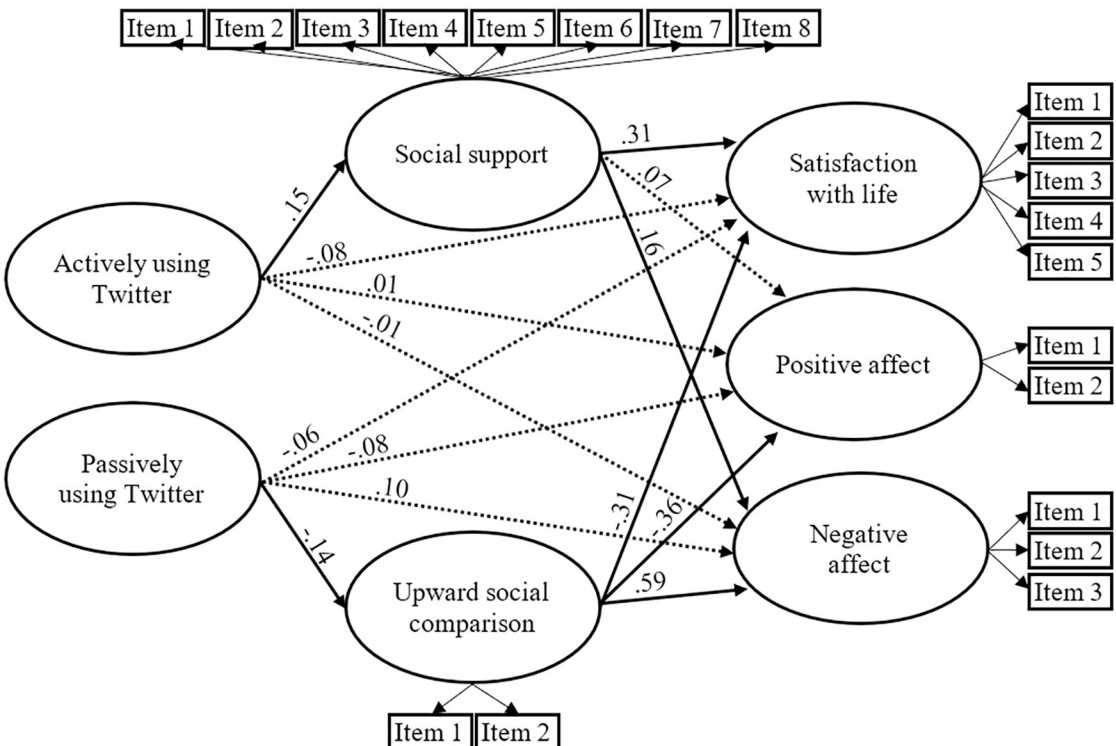

**Fig 3. The estimated standardized parameters of the Twitter model.** Dashed lined indicate nonsignificant paths ($p > .05$). The three components of well-being were controlled–but not displayed—for gender and age: Age was associated with satisfaction with life ($\beta = .26$, $p < .05$), negative affect ($\beta = -.33$, $p < .05$), and positive affect ($\beta = .26$, $p < .05$); women had less positive affect ($\beta = -.30$, $p < .05$), and more negative affect ($\beta = .14$, $p < .05$).

through upward social comparison [20]. This study extends the scope of this conclusion by systematically testing the model to various SNSs (Facebook, Instagram, Twitter, TikTok) within a wider context: the COVID-19 pandemic.

First of all, participants' increase in the use of all SNSs during the quarantine measures strengthens the need to explore the relation between SNSs and well-being. Consistent with Verduyn et al. [20], upward social comparison mediated the negative association between passively using Facebook and well-being. Nevertheless, no relation was found for active Facebook usage and social support (hypothesis 1 partially supported). Instagram showed the opposite relation: social support mediated the positive association between actively using Instagram and well-being (satisfaction with life and negative affect). In contrast to Chae [22], no relation was found for passively using Instagram and upward social comparison (hypothesis 2 partially supported). One surprising outcome is that negative affect were positively related to social support and using actively Facebook. However, in line with Rimé, Bouchat, Paquot and Giglio [40], it is plausible that interacting with others on SNSs elicits emotional reactivation rather than discharge. As a consequence, obtaining social support during the COVID-19 pandemic, a negative and painful event, may increase negative affect. This result is particularly interesting and highlights the role of the socio-emotional context in the relation between SNSs and well-being. As regard to TikTok, no association with well-being, social support or upward social comparison was found (hypothesis 3 not supported). Finally, actively using Twitter was associated with more social support, and passively using Twitter with less upward social comparison.

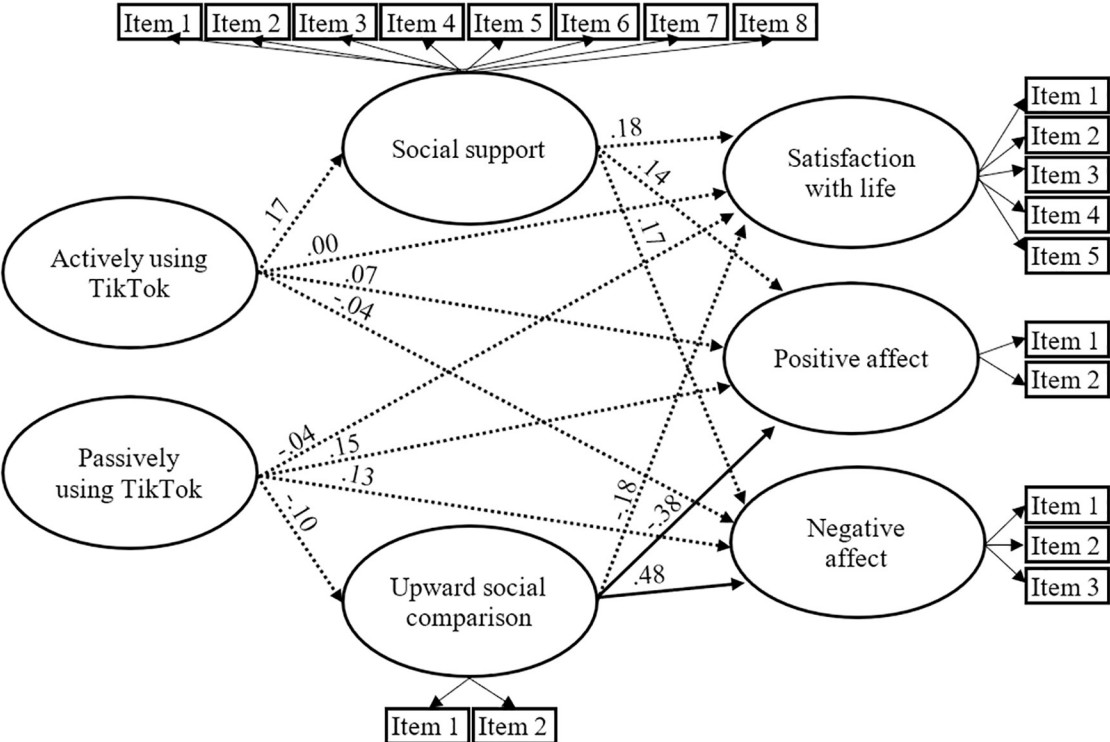

**Fig 4. The estimated standardized parameters of the TikTok model.** Dashed lined indicate nonsignificant paths ($p$ >.05). The three components of well-being were controlled–but not displayed—for gender and age: women had less positive affect ($\beta$ = -.30, $p$ < .05), and more negative affect ($\beta$ = .26, $p$ < .05).

Furthermore, social support mediated the relation between using actively Twitter and satisfaction with life, and upward social comparison mediated the relation between passively using Twitter and negative affect (hypothesis 4 not supported). In other words, our results are fully consistent with those of Chae [22] and demonstrate that, rather than an absence of relation [21], both active and passive usage of Twitter can be positively related to well-being. Which might seem surprising—the negative association between passive usage of Twitter and upward social comparison–may find an explanation in the social context of Twitter. Indeed, previous studies have shown that negative messages are shared faster on Twitter [41] and that popular events on Twitter are associated with negative emotions [42]. Recently, Waterloo, Baumgartner, Peter and Valkenburg [43] showed than negative emotions are perceived as more appropriate on Facebook and Twitter, compared to Instagram. Hence, it is plausible that Twitter's users scrolling through their Twitter news feed and seeing constant bad news from their followers, are more inclined to compare their situation with what they consider to be worse (i.e. downward social comparison), rather than better off (i.e. upward social comparison). Conversely, Facebook is known to be a place for positive self-presentation and impression management [26], which could explain the positive association with upward social comparison.

In that respect, it seems that the model proposed by Verduyn et al. [20] does not stand for every kind of SNSs. Facebook and Instagram use matched partially to the underlying mechanisms, but TikTok use had almost no relation to well-being, and passive usage of Twitter was negatively associated with upward social comparison. The issue is therefore to understand what characteristics and features of SNSs are accountable for these differences. In contrast with Pittman and Reich [21], the findings did not support the architecture of SNSs. Rather, it

seems that users' motivations are more indicative: Twitter and TikTok use during quarantine were not related with social relationships, contrary to Facebook and Instagram. But while Tik-Tok use was only related to entertainment, Twitter use was also related to some purposive values which are considered as a subtype of social support [i.e., informational support, 32]. This is, by the way, fully in line with the literature on motivations to use Facebook, Instagram, Twitter, and TikTok [26–30]. Future studies should further explore how users' motivations affect the relation between SNSs and well-being.

Concerning the specific context of the COVID-19 pandemic, in the words of IJzerman et al. [44] "psychological science is not yet a crisis-ready discipline", and caution should therefore be taken to give recommendations. This single study does not allow to give advice. Maybe conclusions are solely to not systematically promote an overall use of SNSs but rather to distinguish active and passive usages, and to differentiate social network sites due to their specificities.

Finally, the present study is not devoid of limitations. Since participants were recruited via academic mailing lists from social science, the sample is quite biased towards academic people, as well as women (there are a majority of women in social science). This kind of limitation is common in research about social network sites, but we could suspect that this unbalance-sample limits the generalization of the results. Likewise, the small number of participants having a TikTok account, and the fact that all participants were Francophone, highlight the need to replicate the study in other populations. Second, to avoid demotivating respondents, we have limited the questionnaire length. Consequently, passive and active SNS usages have been measured with one item. Although they are considered as separated constructs in the literature [13], we may suspect that the use of single items has increased their association. Future studies should therefore assess specific activities on each SNS. For the same reason, only three kinds of motivation were included. But there is a lot of other reasons to use social network sites, like self-enhancement or self-documentation. Thirdly, as noted by an anonymous reviewer, we think that another good way of assessing our hypotheses would have been to test a model including all social network simultaneously. However, we think that this kind of modeling requires much more participants to draw valid conclusions. Last but not least, this study is cross-sectional, which do not allow to speak in terms of causality or consequences. For example, this study cannot support if people with lower well-being go on SNSs to increase their social support [45]. Future studies should therefore employ longitudinal and experimental designs.

## Conclusions

The current research addresses the complex relation between SNSs and well-being. It extends the literature on passive and active usages by opening the reflection on various kinds of SNS. Passive usage of Facebook was related to social comparison, which, in turn, was associated with lower well-being. Besides, active usage of Instagram was related to social support, which, in turn, was associated with greater satisfaction with life but also negative affect. Regarding Twitter, active usage was also related to social support, which, in turn, was associated with greater satisfaction with life; but passive usage was rather negatively associated with upward social comparison, which, in turn, was associated with more negative affect. In contrast, Tik-Tok use was not associated with well-being. Taken together, this study demonstrates that the differences between SNSs must be considered to truly investigate how SNSs shape human interactions—generalization to every kind of SNS should always be undertaken with caution.

## Acknowledgments

We thank the participants for having dedicated their precious time to our study, especially in these difficult times.

## Author Contributions

**Conceptualization:** Alexandra Masciantonio, David Bourguignon, Pierre Bouchat, Manon Balty, Bernard Rimé.

**Data curation:** Alexandra Masciantonio, Pierre Bouchat.

**Formal analysis:** Alexandra Masciantonio, Pierre Bouchat.

**Investigation:** Alexandra Masciantonio, David Bourguignon, Pierre Bouchat, Manon Balty, Bernard Rimé.

**Methodology:** Alexandra Masciantonio, David Bourguignon, Pierre Bouchat, Manon Balty, Bernard Rimé.

**Project administration:** David Bourguignon, Bernard Rimé.

**Resources:** Alexandra Masciantonio, David Bourguignon, Pierre Bouchat, Bernard Rimé.

**Software:** Alexandra Masciantonio, Pierre Bouchat.

**Validation:** Alexandra Masciantonio.

**Visualization:** Alexandra Masciantonio.

**Writing – original draft:** Alexandra Masciantonio.

**Writing – review & editing:** Alexandra Masciantonio, David Bourguignon, Pierre Bouchat, Bernard Rimé.

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
