## [Decision Letter · Decision Letter 0]

20 Nov 2020

PONE-D-20-27590

Don’t put all social network sites in one basket: Facebook, Instagram, Twitter, TikTok and their relations with well-being during the COVID-19 pandemic.

PLOS ONE

Dear Dr. Masciantonio,

Thank you for submitting your manuscript to PLOS ONE. After careful consideration, we feel that it has merit but does not fully meet PLOS ONE’s publication criteria as it currently stands. Therefore, we invite you to submit a revised version of the manuscript that addresses the points raised during the review process.

Dear authors, 

the manuscript requires several improvements, as highlighted in the reviews. Principally, the dataset needs to be improved with more data and it should be analysed more in details.

Please, follow all the requirements suggested by the review before the re-submission of the paper.

We look forward to receiving your revised manuscript.

Kind regards,

Barbara Guidi

Academic Editor

PLOS ONE

Journal Requirements:

Reviewers' comments:

Reviewer's Responses to Questions

**Comments to the Author**

1. Is the manuscript technically sound, and do the data support the conclusions?

Reviewer #1: Yes

Reviewer #2: No

Reviewer #3: Partly

2. Has the statistical analysis been performed appropriately and rigorously? 

Reviewer #1: Yes

Reviewer #2: Yes

Reviewer #3: No

3. Have the authors made all data underlying the findings in their manuscript fully available?

Reviewer #1: No

Reviewer #2: No

Reviewer #3: Yes

4. Is the manuscript presented in an intelligible fashion and written in standard English?

Reviewer #1: Yes

Reviewer #2: Yes

Reviewer #3: Yes

5. Review Comments to the Author

Reviewer #1: This article deals with an interesting question: the relationship between social network usage and well-being during the COVID-19 pandemics. The article expands the horizon of the review made by Verduyn et al. (2017), by providing an analysis through multiple platforms, based on structural equation modeling. The research design is similar to the one described in Verduyn et al. (2017), with the same two mediators: social support and upward social comparison.

The procedure is correct, and the contribution of the work comes from the results it presents, which might help understand human interactions through social networks. From the methodological point of view, I think the work is not innovative. However, the procedure is sound and the literature review is also correct.

As I flaw, I found the data quite biased towards the female population (line 139): 77% females vs. 23% males, which the authors recognize as a limitation. I wonder if it is also biased towards certain socio-economic segments and age segments, as the cohort was obtained from an academic mailing list. Though this is common in many research works in this area, I think that this type of limitations should be briefly discussed.

In general lines, and despite its limitations, I think that the work can be considered for publication in PLOS ONE.

Minor details:

- This phrase seems contradictory (line 89) "Recently, Chae (21) showed a negative relationship between the use of Twitter, Instagram, LinkedIn and relative well-being through social comparison. Specifically, Instagram and LinkedIn enhanced social comparison, whereas Twitter decreased social comparison." I guess that the first sentence should be neutral: "Recently, Chae (21) showed a negative relationship between the use of Twitter, Instagram, LinkedIn and relative well-being through social comparison. Specifically, ....". Otherwise, it would be interpreted that there it showed a negative relationship between the use of Twitter and relative well-being through social comparison, and I think it was not the case. Please check.

- Line 213: "khi-deux" should read "chi-square".

Finally, I remark that the data used in this research has not been made available with the submission. I understand that the authors will make it public after acceptance, according to the PLOS ONE policy on Data Availability (http://journals.plos.org/plosone/s/data-availability). Data availability is mandatory, and should be checked before publication.

Reviewer #2: In this paper the authors present a study concerning possible correlations of the active/passive usage of social networks and medias and well-being, positive effects and negative effects. The paper seems overall well written, although some typos were found, but there are a number of very important weaknesses listed below.

1) All the key concepts of this paper are not well defined and are extremely vague. I did not find any definition of "well-being", for instance. Is it related to "being healthy"? In this case probably the model is not complex enough to capture the concept of "well-being" because it lacks other important aspects. The same goes for "Satisfaction with life", Positive affects", and "Negative affects". The clarification of these concepts would make the paper clear/easy to understand and technically sound.

2) In Section 2 there is a discussion concerning the 4 sns used for the analyses. A part of the discussion can be easily summarised considering the fact that Facebook is a Social Network (build a network of known people), while the other three are Social Media (media sharing platforms). I agree that the study should be carried considering all the platforms separately, but I somewhat expect some similarities between social media platforms, especially between Instagram and TikTok.

3) The dataset does not seem to be relevant for the study.

- It is made of only 793 people

- The number of women is much higher. In many datasets it was shown that usually the number of females and males on sns are similar or only slightly unbalanced. But not 75% women and 25% men.

- They are all francophone, thus probably living in a specific geographic region

- Participants were recruited using academic mailing lists, thus they are probably all academic people.

- The number of people using Twitter or TikTok seems very low.

The low number of people and all the other very specific features makes me think that probably there is a very strong bias in the dataset.

4) The authors should spend more effort in motivating the methodology. Here are some questions:

- line 154: Why did you use a 7-point scale?

- lines 159-160: does that mean that a user can be both very active and very passive at the same time? And what does "very actively" means? Once per day is "very active"?

- line 172: you use Likert scale extensively in your paper. A relevant citation would increase the quality of the paper. Additionally, why did you use that scale? are there alternatives? Why is Likert the best choice in this scenario?

5) I would spend more effort also in the reorganisation of the contents in the paper: sometimes results (like the McDonald's omega) is shown in the framework presentation section, and part of the framework (lines 210-218) is presented in the result section. You should separate better the framework from the results to help readability.

typos:

line 35: "Analyzes employed", plural of "analysis" is"analyses"

lines 60-64: probably there is some problems with the indentation of the text here

Reviewer #3: The manuscript in question approaches an important problem of how different types of social networks affect subjective well-being of users.

The authors design a thorough survey that includes assessment of such measures as motivation, social support, upward social comparison etc.

Then they use a well established framework of structural equation modeling (SEM) to evaluate direct and indirect effects of active and passive use of social networks on well-being comprised of satisfaction with life, positive affect and negative affect.

1) The paper would benefit from exploratory data analysis: how do the distribution of measures look, what are correlations between them? It is an important first step that can serve as a sanity check when using SEM.

2)

a) Two genders are well represented in this study. It is clear that gender might be an important factor in determining how well-being is derived from the use of social networks, so it should included during modeling.

At the very least it would be interesting to compare the distributions of measures by gender.

b) The gender ratio is imbalanced. Why is it the case? How does it compare to the gender ratio of the recipients of the survey invitation? Does it create a bias? Do those who are not likely to participate in surveys use social networks in the same way? Well, given the gender imbalance of this survey and if there are significant gender differences in SN use, we can hypothesize that those who haven't participated have the inverse gender ratio and so their average motivation and modus operandi might be very different.

3) It appears that the measures were deduced for users who potentially use a mix of social networks. So if someone uses both Facebook and Instagram how do we estimate the fraction that contributes to his motivation for each network?

It seems like a good model should include the use of all networks simultaneously.

4) It would be nice to see a discussion of the impact of the survey invitation being circulated in academic mailing lists and the bias it potentially introduces.

5) Active and passive SN use are treated independently, if I understand correctly. But this is an assumption, and at the very least it should be discussed. It would be interesting to see if the three groups of those who are using actively, passively and both actively and passively have similar distributions of measures.

I conclude that the manuscript is an analysis of the effect social networks on personal well-being based of a large and feature-rich dataset. However, the dataset analysis is incomplete.

This manuscript requires a major revision. Once all the points raised in this review are addressed it can be published in PLOS.

6. PLOS authors have the option to publish the peer review history of their article (what does this mean?). If published, this will include your full peer review and any attached files.

Reviewer #1: No

Reviewer #2: No

Reviewer #3: No

---

## [Author Response · Author response to Decision Letter 0]

2 Dec 2020

We are re-submitting in PLOS One our paper "Don’t put all social network sites in one basket: Facebook, Instagram, Twitter, TikTok, and their relations with well-being during the COVID-19 pandemic.". 

We thank the editor and the reviewers for their time spent carefully reviewing our manuscript, and for their valuable comments. We made sure that each one of the reviewer’s comments has been addressed carefully; we believe that the manuscript has been really improved.

Here are the major revisions made to the manuscript: 

• Regarding the unbalance-sample, dependent variables were controlled for gender and age. This change did not affect the models for Facebook, Instagram and TikTok, but led to substantial revisions in the Twitter’s model. We think that this contribution enhances the scope of our discussion.

• The definition of what is meant by a passive and an active usage of social network sites had been more detailed.

• As requested, we made our date available on an online repository; the DOI necessary to access our data is (anonymized link for blind peer review): https://osf.io/s5mjx/?view_only=b852a4a3eb884b8bb11b83256bee0161. We also made sure that the manuscript meets PLOS ONE's style requirements.

Minor revisions, if not explained, are applied to the manuscript, mostly for correcting typing errors, and for minor modification along with revisions explained in this document. 

The responses to all the reviewer’s comments are detailed in the "Response to reviewers" file. 

Please let us know if you still have any questions or concerns about the manuscript. We will be happy to address them. 

Sincerely, 

The authors of paper PONE-D-20-27590.

---

## [Decision Letter · Decision Letter 1]

21 Dec 2020

PONE-D-20-27590R1

Don’t put all social network sites in one basket: Facebook, Instagram, Twitter, TikTok, and their relations with well-being during the COVID-19 pandemic.

PLOS ONE

Dear Dr. Masciantonio,

Thank you for submitting your manuscript to PLOS ONE. After careful consideration, we feel that it has merit but does not fully meet PLOS ONE’s publication criteria as it currently stands. Therefore, we invite you to submit a revised version of the manuscript that addresses the points raised during the review process.

Reviewers highlighted that the manuscript needs minor revisions in order to be accepted as a possible publication. Please revise the paper by following the suggestions given by the reviewers.

We look forward to receiving your revised manuscript.

Kind regards,

Barbara Guidi

Academic Editor

PLOS ONE

Reviewers' comments:

Reviewer's Responses to Questions

**Comments to the Author**

1. If the authors have adequately addressed your comments raised in a previous round of review and you feel that this manuscript is now acceptable for publication, you may indicate that here to bypass the “Comments to the Author” section, enter your conflict of interest statement in the “Confidential to Editor” section, and submit your "Accept" recommendation.

Reviewer #1: All comments have been addressed

Reviewer #2: (No Response)

Reviewer #3: (No Response)

2. Is the manuscript technically sound, and do the data support the conclusions?

Reviewer #1: Yes

Reviewer #2: Partly

Reviewer #3: Partly

3. Has the statistical analysis been performed appropriately and rigorously? 

Reviewer #1: Yes

Reviewer #2: Yes

Reviewer #3: Yes

4. Have the authors made all data underlying the findings in their manuscript fully available?

Reviewer #1: Yes

Reviewer #2: Yes

Reviewer #3: Yes

5. Is the manuscript presented in an intelligible fashion and written in standard English?

Reviewer #1: Yes

Reviewer #2: Yes

Reviewer #3: Yes

6. Review Comments to the Author

Reviewer #1: The authors have addressed all the remarks and the quality of the manuscript has been improved. In particular, they have correctly discussed the limitations, clarified some relevant definitions for their work, and made their dataset available.

I consider that the manuscript can be accepted for publication in PLOS ONE.

Minor note: I observe that the authors have changed the subsection title "Measures" into "Materials" on line 164. I think that "Measures" might be more appropriate and standard in the field.

Reviewer #2: Authors put an extraordinary effort to revise and improve the paper. I have just a few follow-up points:

- line 33: you claim that 1008 people took the test, but on line 146 you claim that 1004 people took the test. Please, put the correct number in the paper.

- lines 170-179: now it's much more clearer what "actively" and "passively means", but I still wonder whether it was a good idea to keep these two factors separated. In this way one can be "non active" and "non passive" at the same time which does not make much sense (or am I still missing something?). Additionally, if I got it right, they are mutually exclusive activities: if I am scrolling through posts, I am not creating posts at the very same time. I may spend equal time in active and passive behaviour, and that's why I think that a single indicator is better here. Can you please motivate in the paper why you needed two different indicators? You also asked the participants their "overall SNS use", but I don't see it used in the paper, why?

- concerning your data, I understand your limitations, and that's fine, but you should state more clearly in the abstract and the introduction sections that this is a preliminary work.

Reviewer #3: I would like to thank the authors for such a quick and thorough revision.

From my point of view all of the suggested modification have been implemented except for item 1).

The descriptive statistics and correlation table following the link

https://osf.io/s5mjx/?view_only=b852a4a3eb884b8bb11b83256bee0161

shed little light.

I would still recommend trying to visualize the exploratory data analysis (EDA) in terms of histograms and KDE plots of PDFs. See, for example, https://seaborn.pydata.org/generated/seaborn.pairplot.html

One could separate the group of actively using social networks per network by its median into two and plot PDFs for positive affect, negative affect etc questions, or directly use x-y scatter plots with with KDE.

I would encourage to perform such EDA for all available variables, including age, gender etc

With EDA plots the reader would be prepared and actually expect the result. The benefits of EDA include dataset consistency check and motivation of the model: is the input dataset biased and what conclusion should we expect from the model?

7. PLOS authors have the option to publish the peer review history of their article (what does this mean?). If published, this will include your full peer review and any attached files.

Reviewer #1: No

Reviewer #2: No

Reviewer #3: No

---

## [Author Response · Author response to Decision Letter 1]

20 Jan 2021

We responded to each point raised by the academic editor and reviewers in the separate file labeled 'Response to Reviewers'.

---

## [Decision Letter · Decision Letter 2]

9 Feb 2021

PONE-D-20-27590R2

Don’t put all social network sites in one basket: Facebook, Instagram, Twitter, TikTok, and their relations with well-being during the COVID-19 pandemic.

PLOS ONE

Dear Dr. Masciantonio,

Thank you for submitting your manuscript to PLOS ONE. After careful consideration, we feel that it has merit but does not fully meet PLOS ONE’s publication criteria as it currently stands. Therefore, we invite you to submit a revised version of the manuscript that addresses the points raised during the review process.

The paper should be revised. Please follow the MINOR SUGGESTIONS given by the reviewers.

We look forward to receiving your revised manuscript.

Kind regards,

Barbara Guidi

Academic Editor

PLOS ONE

Reviewers' comments:

Reviewer's Responses to Questions

**Comments to the Author**

1. If the authors have adequately addressed your comments raised in a previous round of review and you feel that this manuscript is now acceptable for publication, you may indicate that here to bypass the “Comments to the Author” section, enter your conflict of interest statement in the “Confidential to Editor” section, and submit your "Accept" recommendation.

Reviewer #1: All comments have been addressed

Reviewer #2: All comments have been addressed

Reviewer #3: All comments have been addressed

2. Is the manuscript technically sound, and do the data support the conclusions?

Reviewer #1: Yes

Reviewer #2: Partly

Reviewer #3: Partly

3. Has the statistical analysis been performed appropriately and rigorously? 

Reviewer #1: Yes

Reviewer #2: Yes

Reviewer #3: Yes

4. Have the authors made all data underlying the findings in their manuscript fully available?

Reviewer #1: Yes

Reviewer #2: Yes

Reviewer #3: Yes

5. Is the manuscript presented in an intelligible fashion and written in standard English?

Reviewer #1: Yes

Reviewer #2: Yes

Reviewer #3: Yes

6. Review Comments to the Author

Reviewer #1: I appreciate the data exploration analysis that the authors have included in the current revision.

As the authors have satisfactorily addressed the comments, I consider that the manuscript can be accepted for publication in PLOS ONE.

Reviewer #2: (No Response)

Reviewer #3: The exploratory data analysis was nominally performed, however, it is completely detached from the rest the analysis (and not very informative).

Usually it serves the purpose of motivating further analysis using more advanced techniques.

For instance, in lines 266-268,"Age was associated with satisfaction ... with life negative affect ... and positive affect"

The significance and the signs of effects should be manifest in EDA.

I believe any reader would appreciate an announcement of strong correlation in the prelude and a following confirmation by a stronger method.

7. PLOS authors have the option to publish the peer review history of their article (what does this mean?). If published, this will include your full peer review and any attached files.

Reviewer #1: No

Reviewer #2: No

Reviewer #3: No

---

## [Author Response · Author response to Decision Letter 2]

15 Feb 2021

We are re-submitting in PLOS One our paper "Don’t put all social network sites in one basket: Facebook, Instagram, Twitter, TikTok, and their relations with well-being during the COVID-19 pandemic.". 

We thank again the editor and the reviewers for the time they spent reviewing our manuscript. 

The detailed responses to all the reviewer’s comments are available in the "Response to Reviewers" file. 

Sincerely, 

The authors of paper PONE-D-20-27590.

---

## [Decision Letter · Decision Letter 3]

26 Feb 2021

Don’t put all social network sites in one basket: Facebook, Instagram, Twitter, TikTok, and their relations with well-being during the COVID-19 pandemic.

PONE-D-20-27590R3

Dear Dr. Masciantonio,

We’re pleased to inform you that your manuscript has been judged scientifically suitable for publication and will be formally accepted for publication once it meets all outstanding technical requirements.

Kind regards,

Barbara Guidi

Academic Editor

PLOS ONE

Additional Editor Comments (optional):

Reviewers' comments:

Reviewer's Responses to Questions

**Comments to the Author**

1. If the authors have adequately addressed your comments raised in a previous round of review and you feel that this manuscript is now acceptable for publication, you may indicate that here to bypass the “Comments to the Author” section, enter your conflict of interest statement in the “Confidential to Editor” section, and submit your "Accept" recommendation.

Reviewer #1: All comments have been addressed

Reviewer #2: All comments have been addressed

2. Is the manuscript technically sound, and do the data support the conclusions?

Reviewer #1: Yes

Reviewer #2: Partly

3. Has the statistical analysis been performed appropriately and rigorously? 

Reviewer #1: Yes

Reviewer #2: Yes

4. Have the authors made all data underlying the findings in their manuscript fully available?

Reviewer #1: Yes

Reviewer #2: Yes

5. Is the manuscript presented in an intelligible fashion and written in standard English?

Reviewer #1: Yes

Reviewer #2: Yes

6. Review Comments to the Author

Reviewer #1: All comments have been addressed and the exploratory analysis has been expanded. I consider that the article can be accepted for publication in PLOS ONE.

Reviewer #2: (No Response)

7. PLOS authors have the option to publish the peer review history of their article (what does this mean?). If published, this will include your full peer review and any attached files.

Reviewer #1: No

Reviewer #2: No

---

## [Editor Report · Acceptance letter]

3 Mar 2021

PONE-D-20-27590R3 

Don’t put all social network sites in one basket: Facebook, Instagram, Twitter, TikTok, and their relations with well-being during the COVID-19 pandemic. 

Dear Dr. Masciantonio:

I'm pleased to inform you that your manuscript has been deemed suitable for publication in PLOS ONE. Congratulations! Your manuscript is now with our production department. 

Kind regards, 

on behalf of

Dr. Barbara Guidi 

Academic Editor

PLOS ONE